# Effect of Silicon Wafer Surface Stains on Copper-Assisted Chemical Etching

Liang Ma [1], Xiuhua Chen [2], Chenggui Tang [3], Shaoyuan Li [1,4,*], Fengshuo Xi [1,*], Huayan Lan [4], Wenhui Ma [1,4] and Yuanchih Chang [5]

1 Faculty of Metallurgical and Energy Engineering/Key Laboratory of Complex Nonferrous Metal Resources Clean Utilization, Kunming University of Science and Technology, Kunming 650093, China
2 School of Materials and Energy, Yunnan University, Kunming 650091, China
3 QuJing LONGi Silicon Materials Co., Ltd., Qujing 655100, China
4 Yunnan Provincial Rural Energy Engineering Key Laboratory, School of Science and Technology, Pu'er University, Pu'er 665000, China
5 Australian Centre for Advanced Photovoltaics, School of Photovoltaic and Renewable Energy Engineering, University of New South Wales, Sydney 2052, Australia
* Correspondence: lsy415808550@163.com (S.L.); fengshuoxi@126.com (F.X.)

**Abstract:** Silicon wafer slicing is a crucial process during solar cell fabrication, but it often stains the silicon wafer surface. Thus, this work systematically investigated the composition, source, and cleaning method of typical white spot stains on silicon wafer surfaces. The EDS and XPS results showed that the white spot stains contained $CaCO_3$ and $SiO_2$ that were consistent with the filler components in sticky silicon ingot glue. The effects of stains on copper deposition and copper-assisted chemical etching were studied. White spot stains remained attached to the silicon surface after deposition and etching. These stains affected the uniform deposition of copper particles on the surface of the silicon wafer and also impeded the catalytic etching of copper particles. Finally, KOH solution was combined with an ultrasonic field to remove surface stains from the silicon wafer. This study provides important guidance for the removal of silicon wafer contaminants to fabricate high-efficiency solar cells.

**Keywords:** silicon wafer; white spot stains; silicon wafer texturization; silicon wafer cleaning

## 1. Introduction

A stable supply of energy supports human survival and sustainable social development. Since the Industrial Revolution, fossil energies such as coal and petroleum have been consumed in large quantities, but these have also produced environmental pollution such as massive greenhouse gas emissions, causing global warming and climate change. Therefore, renewable energy sources such as photovoltaics and wind power have been rapidly developed. The global installed photovoltaic capacity increased by 21 times from 2010 to 2021 and reached 965 GW at the end of 2021. It is expected that the installed photovoltaic capacity will enter the TW regime in 2022 and reach almost 5 TW by 2030 [1–3].

Silicon materials are the market leader for the entire photovoltaic industry [4]. In the silicon-based photovoltaics industry chain, the fabrication of silicon wafers is the most basic step. Diamond slicing is the main silicon wafer slicing technology in which high-hardness diamonds on steel wire are used to slice silicon into thin sheets by high-speed linear friction [5,6]. During silicon wafer fabrication, manufacturing equipment, auxiliary materials, and negligent operations may produce silicon wafer surface stains [7–9]. During the slicing of silicon wafers, due to the repeated rubbing of diamond wires and silicon wafers, a large amount of brittle damage and plastic damage occurs on the silicon wafer surface. Due to this, the density of dangling bonds must be very high, and these dangling bonds have high reactivity and strong activity. The damaged surface often adsorbs various

surrounding molecules or atoms that stain the surface of the silicon wafer [10–12]. These stains are mainly produced via physical adsorption and chemical adsorption. Physically adsorbed stains are mainly formed due to silicon waste, dust particles, glue, and other substances that do not react with silicon wafers. Silicon wafers combine with this type of stain through van der Waals forces. Chemical stains are mainly formed due to contaminants in the processing equipment and due to the shedding of small metal particles and metal ions in the slicing fluid. This type of stain combines with silicon wafers via chemical bonds such as ionic bonds and covalent bonds [13–15].

Stains on the surface of the silicon wafer affect the texture and appearance of the silicon wafer. Some stains may spread to the Si substrate to form carrier recombination centers that decrease a device's performance and reliability [16–18]. There can be many forms of stains on the surface of silicon wafers. Fumitoshi Sugimoto et al. studied the adsorption behavior of organic contaminants on the surface of silicon wafers and found that low-boiling organic contaminants tended to be adsorbed immediately, but high-boiling organic contaminants required a longer exposure time [19]. Sunho Choi et al. found that red-colored contaminants on the surface of a silicon wafer were derived from copper-plated diamond wire. By analyzing particles in the slurry before and after slicing, it was found that the pulp contained a large amount of super-small Cu, SiC, and Si particles that increased the slurry viscosity. By providing additional coolant to dilute the slurry, Cu contamination was eliminated [20]. M. Watanabe et al. analyzed the mechanism by which watermarks formed on the surface of a silicon wafer. They formed when silicon oxide (possibly $SiO_2$) under the water droplets on the surface of the silicon wafer dissolved in the water droplets. After the water droplets dried, $H_2SiO_3$ remained on the surface [21].

Various cleaning methods have been used to remove stains from silicon wafer surfaces; for example, in the brush cleaning method, a sponge brush made of polyvinyl alcohol is used to press on the silicon wafer and clean the particles on the silicon wafer using a rotating sponge-like brush. In the two-fluid spray cleaning method, a high-pressure gas fluid is used to remove stains from the surface. During laser cleaning technology, which uses the characteristics of the stain with strong laser absorption, stains instantaneous absorb a large amount of laser energy, forcing the separation of surface stains and silicon substrate. In the RCA cleaning method, silicon wafers are sequentially placed in a chemical reagent, such as a mixture of $H_2SO_4$ and $H_2O_2$, HF, a combination of $NH_3 \cdot H_2O$ and $H_2O_2$, or a mixture solution of HF and $H_2O_2$ [22–25]. These cleaning methods have good effects, but the cleaning equipment is expensive.

Currently, most stains are easily removed during the silicon wafer cleaning process. However, white spot stains remain on the silicon wafer surface after cleaning, and there have been few detailed studies of these stubborn stains. Silicon wafers need to be textured to reduce surface reflectance to improve photoelectric conversion efficiency. Copper-assisted chemical etching has received widespread attention due to its high efficiency and low cost. The inverted pyramid structure generated by etching has an excellent light-trapping ability [26–28], but there have been no studies on how stains affect this method.

In this paper, the chemical composition of white spot stains on a silicon wafer surface was characterized by X-ray energy-dispersive spectroscopy (EDS) and X-ray photoelectron spectroscopy (XPS). To determine the origin of the white spot stains, copper-assisted chemical etching was used to etch white spot stains on the silicon wafer on which copper was deposited via chemical deposition. The effects of the contaminant on etching and deposition were studied. Finally, the silicon wafer was cleaned by ultrasonication in KOH solution for 5 min.

## 2. Materials and Methods

### 2.1. Testing and Analysis of White-Spot-Stained Silicon Wafers

White stains on a silicon wafer are often encountered during the production of solar cells. The size of the silicon wafer was 210 mm × 210 mm, and n type crystalline silicon wafers had a thickness of 150 ± 5 µm. The resistivity was 1–3 Ω. Plastic board, sticky

silicon ingot glue, and diamond wire were obtained from the enterprise that produced the solar cells. A tungsten filament scanning electron microscope equipped with Oxford EDS (SEM, TESCAN VEGA3, Brno, Czech Republic) was used to characterize the morphology and elemental distribution of white spot stains, plastic board, sticky silicon ingot glue, and diamond wire in detail. The surface state of white spots was analyzed using X-ray photoelectric spectroscopy (XPS, K-Alpha+, Thermo Fisher Scientific, Oxford, UK).

### 2.2. Deposition and Etching

The etching solution consisted of 2.76 M HF, 1.2 M $H_2O_2$, 0.05 M $Cu(NO_3)_2$, and deionized (DI) water. This was placed in a Teflon beaker and heated in a water bath to 40 °C. The white-spot-stained silicon wafer was put into the etching solution and reacted for 7 min. After etching, the samples were thoroughly rinsed with distilled water and dried using nitrogen. The deposition solution consisted of 2.76 M HF, 0.05 M $Cu(NO_3)_2$, and DI water. It was placed in a Teflon beaker at room temperature. A stained silicon wafer was placed into the deposition solution and reacted for 10 s. After deposition, the sample was thoroughly rinsed with distilled water and dried using nitrogen. After the experiment, the morphology of white spot stains before and after etching and deposition was studied using SEM.

### 2.3. Wafer Cleaning

The cleaning solution consisted of KOH (10 wt%). At room temperature, the stained silicon wafer was placed into the cleaning solution with a 200 W and 53 kHz ultrasonic field for cleaning. Stain shedding was observed with a metallographic microscope every minute of cleaning.

### 3. Results and Discussion

Figure 1a shows a photograph of the white-spot-stained silicon wafer. Visible white spot stains are shown in the three yellow boxes in the photograph. The irregular morphology of a white spot stain was observed by SEM, as shown in Figure 1b. The spot was about 144 μm in length and 155 μm in width. The surface of the material was distributed with different-sized highlights. EDS element maps are shown in Figure 1c–f, which show that the white spot stain contained C, Ca, O, and Si elements. C was present in the highest amount, so it is speculated that the white spot stains may be organic matter.

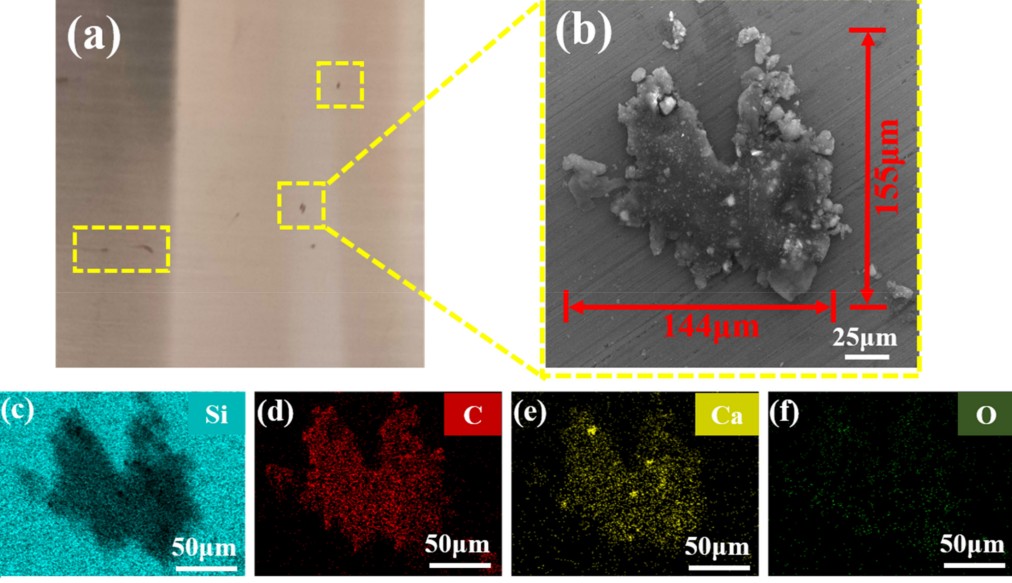

**Figure 1.** (**a**) Photograph of white-spot-stained silicon wafers, (**b**) SEM image of white-spot-stained silicon wafers, (**c**) elemental distribution of Si, (**d**) C, (**e**) Ca, (**f**) and O.

XPS was used to further analyze the valence states and composition of the white spot stains. The XPS spectrum in Figure 2a shows that the white spot stains mainly contained C, Ca, and O, and a small amount of Si. Based on the XPS spectrum in 2a, Ca 2p was deconvoluted into two peaks. The peaks of 347.07 eV and 350.6 eV correspond to Ca $2p_{3/2}$ and Ca $2p_{1/2}$, respectively (Figure 2b) [29]. The peaks at 531 eV in the O 1s spectrum and 347.07 eV and 350.6 eV in the Ca 2p spectrum show that the white spot stains contained CaCO3 particles (Figure 2d). The C 1s spectrum of the white spot stains was decomposed into three components: the peak of C-C was located at 284.7 eV, the peak of C-O-C was at 286.4 eV, and the peak of O-C=O (carbon in carbonate) was located at 289 eV (Figure 2c) [30]. The Si 2p spectrum presented two characteristic peaks at 99 eV and 103.3 eV, which corresponded to monatomic silicon and $SiO_2$, respectively (Figure 2e) [31]. During the production of some polymer materials, $CaCO_3$, $SiO_2$, and other inorganic fillers are often added to glues, rubbers, and other polymeric materials to ensure their mechanical properties and flame retardancy. The results further confirmed that the white spot stains emanated from organic polymer components [32,33].

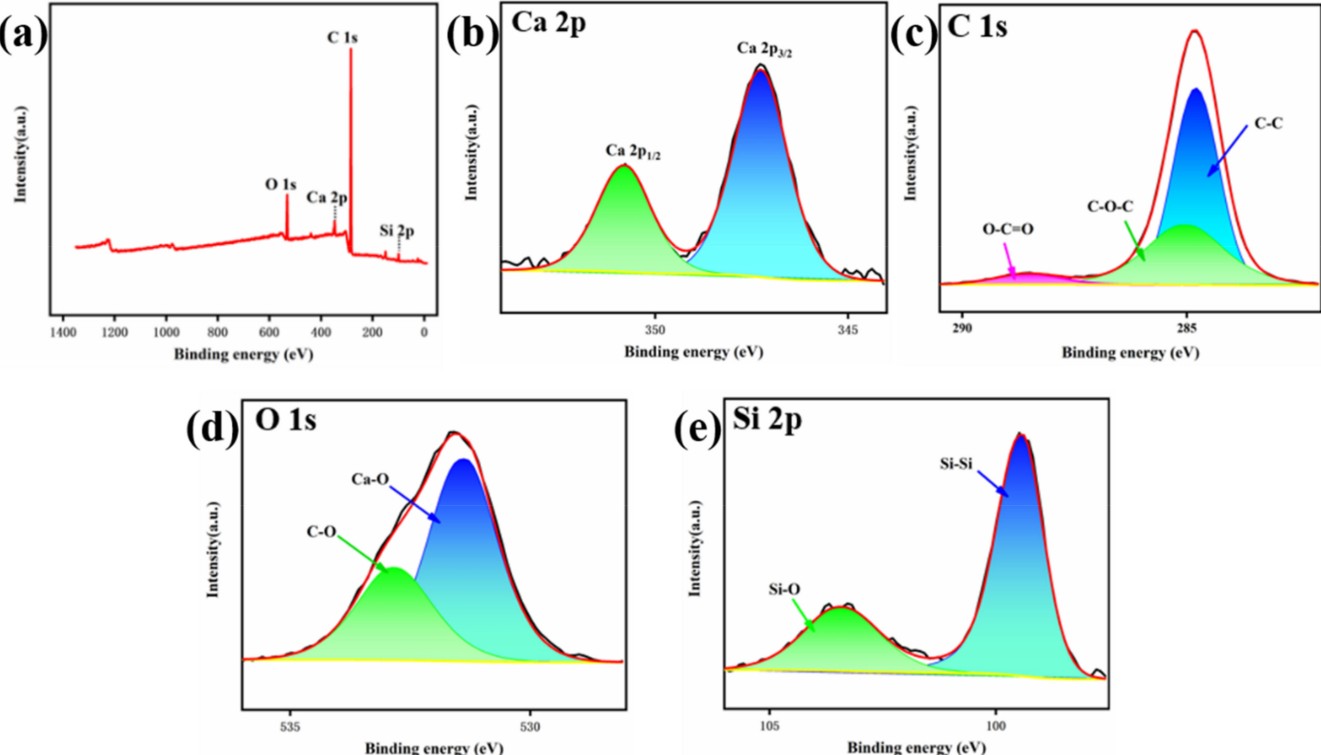

**Figure 2.** (**a**) Full spectrum of XPS spectrum with white spot stain, (**b**–**e**) are the characteristic absorption spectra of Ca2p, C1s, O1s and Si2p.

During the slicing of the silicon wafer, the surface of the silicon wafer will only contact the diamond wire. Therefore, the diamond wire was characterized using EDS, to determine the origin of the stain. Figure 3a shows a new diamond wire, where the surface of the line is covered with diamonds of varying sizes. In comparison with Figure 3a, the diamond on the surface of the line in Figure 3b has been worn to a flat structure, and some diamonds have fallen off. This indicates that it is a used diamond wire. There was a circular protrusion with a diameter of about 20 μm on the surface of the diamond wire. The EDS analysis of the substance is shown in Figure 3b, which shows that it contained C, Ca, O, and Si. However, the diamond wire was nickel plated, and the main elements were Fe, Ni, and C. This means that the substance was not derived from the diamond wire. The elements in this substance are consistent with the elements in the white spot stain, meaning that it is also a polymeric material.

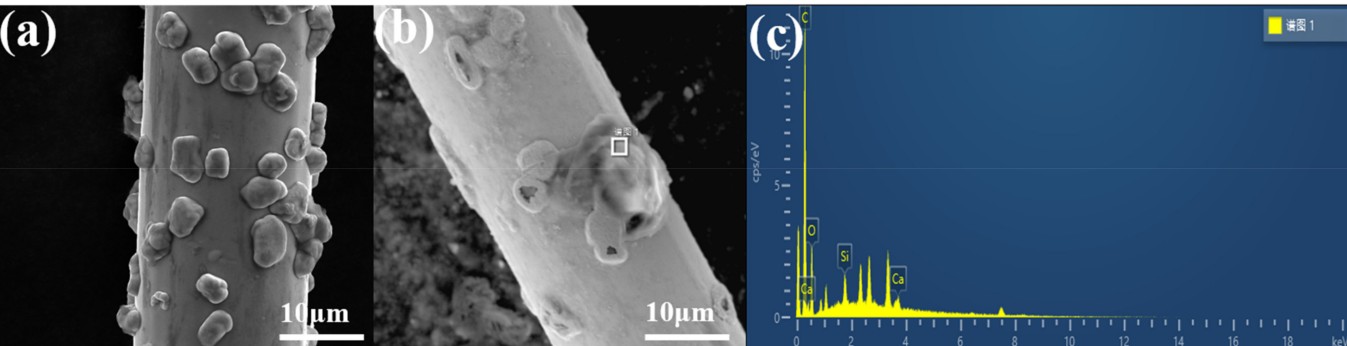

**Figure 3.** (**a**) SEM image of fresh diamond wire, (**b**) SEM image of waste diamond wire, (**c**) EDS spectrum of waste diamond wire surface substances.

To determine where the substance on the diamond wire was introduced, the entire slicing process was analyzed. When slicing silicon wafers, sticky silicon ingot glue is used to hold the silicon ingots and plastic board together. The only polymeric materials that can come into contact with the diamond wire are those found in the sticky silicon ingot glue and the plastic board. The substances in these two elements were analyzed by EDS. Figure 4a shows an SEM image of the plastic sheet covered with slice marks caused by diamond wire slicing. The element maps in Figure 4b,c show that it only contained C and O, indicating that the substance was not introduced by the plastic board. Figure 5a shows an SEM image of sticky silicon ingot glue, in which several areas in the figure are brighter than others. The element map in Figure 5b–e shows that it contained C, Ca, O, and Si. The bright spots in the SEM image are Ca. Sticky silicon ingot glue contains the same elements as the white spot stain and diamond wire, indicating that the substance causing the stains emanated from the sticky silicon ingot glue.

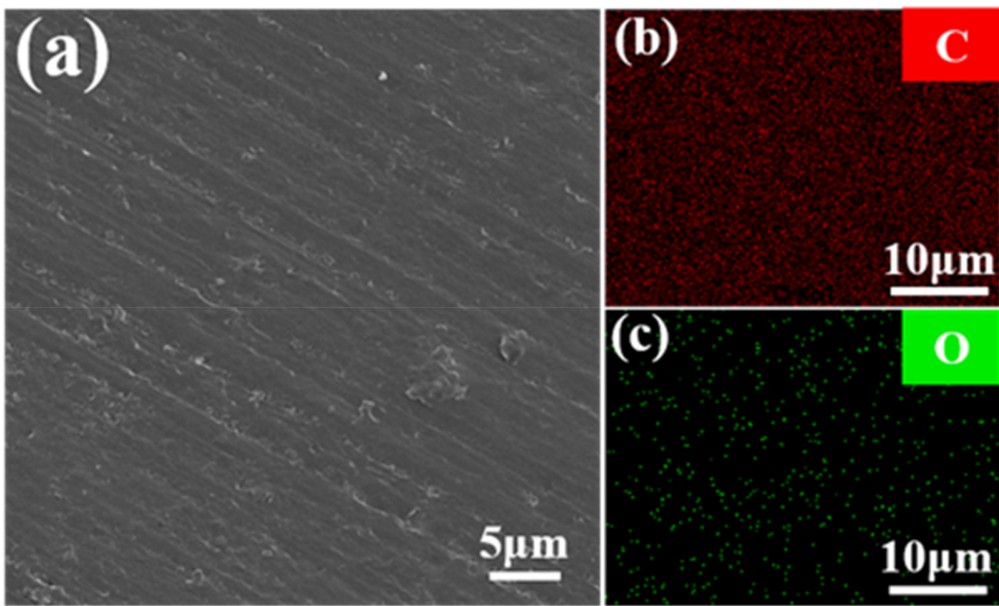

**Figure 4.** (**a**) SEM image of plastic board, (**b**) elemental distribution of C and (**c**) O.

The process of restoring the white spot stains is shown in Figure 6. Yellow is the salver, red is sticky silicon ingot glue, blue is a plastic board, green is the driver pulley, and gray is a silicon wafer. Before slicing, the salver, plastic board, and silicon ingot needed to be bonded using sticky silicon ingot glue. After the glue solidified, the silicon ingot bonded with the salver and plastic board was installed on a diamond cutting machine for cutting. During the slicing process, the diamond wire moved at high speed under the action of the

drive pulley, and the diamond wire was fed vertically to the silicon ingot. The diamond wire threaded stress on the silicon ingot, and the diamonds were driven by the wire thread and ground the silicon ingot, thus slicing the silicon ingot. Due to uneven stress at each point of the wire, a line bow phenomenon occurred [34] in which the outer slicing depth was greater than the inner slicing depth. Therefore, the outer side of the silicon ingot was sliced before the inner side. Ensuring that the diamond wire had enough slicing depth was critical to ensuring that the silicon ingot was fully sliced. A plastic board was added in the middle of the silicon ingot and crystal bead. When the diamond wire sliced the plastic board, the silicon ingot was sliced through, and the diamond wire came into contact with the sticky silicon ingot glue. As a result, a large amount of sticky silicon ingot glue adhered to the diamond wire surface, so when the diamond wire sliced the next batch of silicon ingots, the glue on the diamond line rubbed against the silicon wafer and adhered to the silicon wafer, thus forming stains.

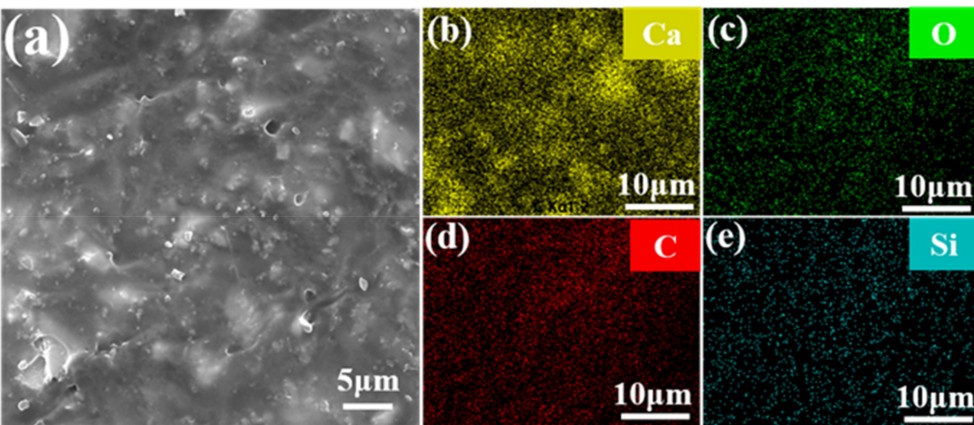

**Figure 5.** (**a**) SEM image of glue, (**b**) elemental distribution of Ca, (**c**) O, (**d**) C, and (**e**) Si.

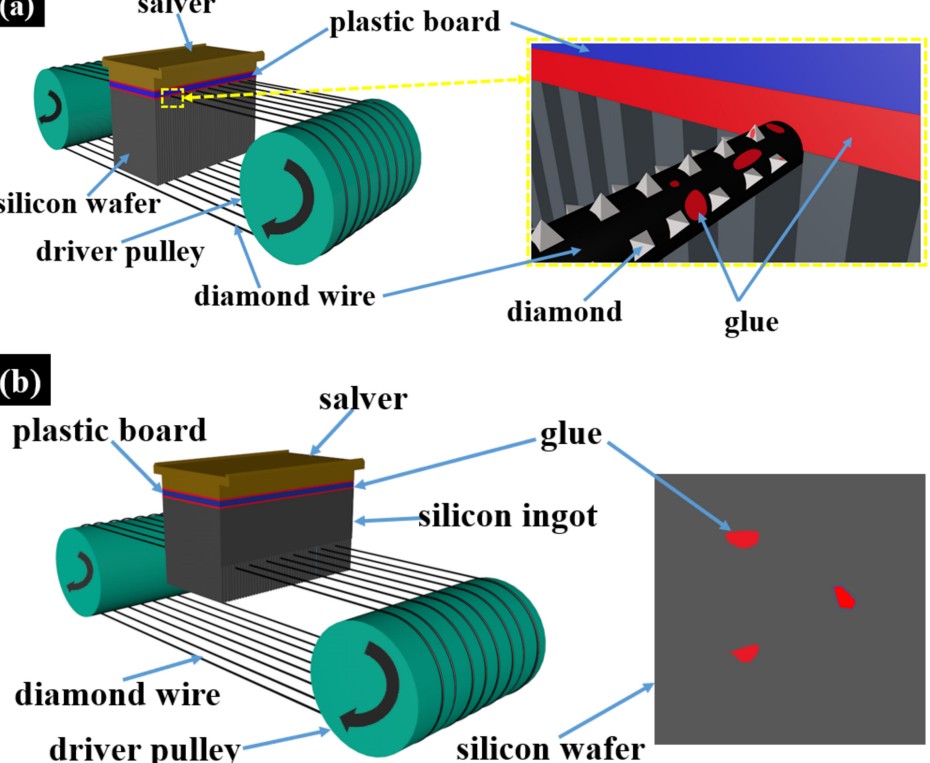

**Figure 6.** (**a**) Schematic diagram of diamond wire attachment stains, (**b**) schematic diagram of adhesion stains of the silicon wafer.

Figure 7a,b show the SEM diagram before and after the deposition of the silicon wafer in HF/Cu(NO$_3$)$_2$ solution. Due to the difference in electronegativity between Si and copper ion (Cu$^{2+}$), the copper ion in the solution can acquire electrons from the wafer surface and form Cu particles; at this point, silicon is oxidized into SiO$_2$, and the resulting SiO$_2$ is dissolved by HF in the solution and the fresh Si is exposed again. When the reduction of copper ions and oxidation of silicon occur periodically, more and more copper particles are deposited on the silicon wafer surface. Figure 7b shows that stain edges near the red dotted line were darker than those in the area outside the red dotted line. Compared with before deposition, the area outside the red dotted line was much brighter, showing that this area was deposited with copper. There were no significant differences in the stain morphology before and after deposition. The stain did not dissolve in the deposition solution. The EDS results show that the distribution of copper outside the red dotted line was denser than the distribution within the red dotted line. There were almost no copper deposits on the surface of the stain. At the edges of the red dotted line, only a very small amount of copper was deposited because CaCO$_3$ in the stain reacted with HF in the deposition solution via the equation 2HF + CaCO$_3$ → CaF$_2$↓ + CO$_2$↑ + H$_2$O. The reaction generated CaF$_2$ precipitate, resulting in the rapid reduction of F$^-$ around the stain. The pH increased, and the conductivity of the solution decreased. Therefore, F$^-$ in the area outside the red dotted line was not added to the reaction interface outside the red solid line. Thus, concentration polarization occurred [35]. F$^-$ in the HF solution easily reacted with Si atoms to form Si-F bonds, which greatly reduced the Si-Si bond strength. This made it easier for Cu$^{2+}$ ions to obtain electrons and be reduced to elemental copper. The F$^-$ concentration around the stain decreased, which decreased the rate of Cu$^{2+}$ deposition, and resulted in uneven copper deposits [36].

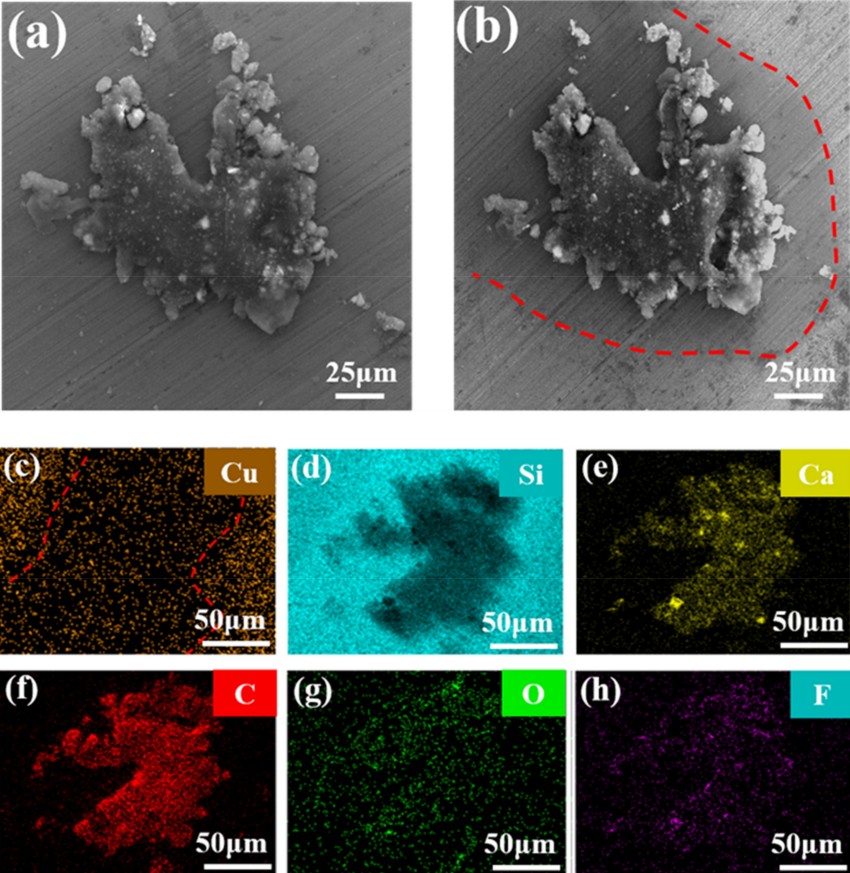

**Figure 7.** (**a**) SEM image of white-spot-stained silicon wafer before and (**b**) after deposition, (**c**) elemental distribution of Cu, (**d**) Si, (**e**) Ca, (**f**) C, (**g**) O, and (**h**) F. (EDS maps in (**c**–**h**) were taken after deposition).

Figure 8 shows an SEM image of a white-spot-stained silicon wafer etched in HF/Cu(NO$_3$)$_2$/H$_2$O$_2$ solution at 40 °C for 7 min. There was oval-shaped material at the center of the picture, and the EDS results showed that the material was a stain adhered to the surface of the silicon wafer. After reacting for 7 min in a strongly oxidizing solution, the stain did not dissolve, and many nanopores were formed by the etching solution in areas that were not covered by the stain; cut marks are still visible. The silicon wafer was etched after KOH treatment, and a dense and uniform inverted pyramidal structure was formed on the silicon wafer surface; cut marks were almost invisible. This was because when slicing silicon wafers, the diamond on the wire exerted a very strong force on the silicon ingot, which caused the dislocation of regularly arranged silicon atoms and formed an amorphous silicon layer. The presence of amorphous silicon layers slowed down the etching rate of the silicon wafers so that only a small amount of the silicon wafer was etched. KOH removed the amorphous silicon layer, which facilitated the deposition of copper particles and improved the etching results [37,38].

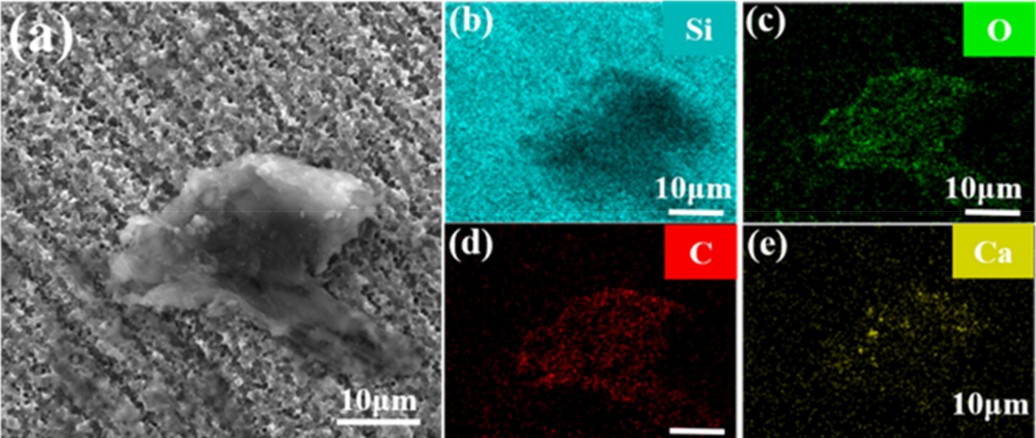

**Figure 8.** (**a**) SEM image of white-spot-stained silicon wafer after etching, (**b**) elemental distribution of Si, (**c**) O, (**d**) C, and (**e**) Ca.

During the etching of silicon wafers using copper-assisted chemical etching, the principle of the etching reaction can be described in terms of the following electrochemical reactions:

$$\text{Cathodic reaction: } Cu^{2+} + 2e^- \rightarrow Cu(s) \tag{1}$$

$$H_2O_2 + 2H^+ \rightarrow 2H_2O + 2h^+ \tag{2}$$

$$\text{Anode reaction: } Si + 6HF + nh^+ \rightarrow H_2SiF_6 + nH^+ + [\frac{4-n}{2}]H_2 \uparrow \tag{3}$$

Based on the above reactions, it can be seen that the main chemical reaction in the HF/Cu(NO$_3$)$_2$/H$_2$O$_2$ etching system is based on the inter conversion of Cu$^{2+}$ and Cu. Due to the difference in electronegativity of Si and Cu$^{2+}$ in solution, the Cu$^{2+}$ near the surface of the silicon wafer gains electrons from the valence bands of silicon and is reduced to Cu nanoparticles that are deposited on the surface of silicon wafer. The copper nanoparticles deposited on the silicon wafer surface increase the electronegativity difference between the surface and Cu$^{2+}$ in the solution, which further promotes the migration of copper ions to the silicon surface and causes a continuous reduction reaction. Under the catalysis of nano-copper particles, H$_2$O$_2$ in the solution is also involved in the cathodic reaction, which plays an important role in the dynamic equilibrium of the whole reaction. H$_2$O$_2$ oxidizes the deposited copper nanoparticles and forms Cu$^{2+}$ ions due to its higher electrode potential and strong oxidability, so that a dynamic equilibrium between the Cu particle formation and dissolution can be reached. In addition, because the redox potential of H$_2$O$_2$ is also above the valence band of silicon, it means that H$_2$O$_2$ can also inject holes into

the valence band of silicon through its own decomposition, and then silicon is oxidized into silicon dioxide and the resulting silicon dioxide is dissolved by HF in solution as a water-soluble compound [39].

The formation of the silicon inverted pyramidal shape is mainly due to the fact that copper-assisted chemical etching is anisotropic etching. During the above electrochemical reaction, the deposition rate of Cu particles in the cathode reaction is related to the rate of electron capture from the surface of the silicon wafer. In addition to the effect of copper ions and $H_2O_2$ in the solution, the loss of electrons on the silicon surface is related to crystal orientation. Consequently, crystal orientation will affect the deposition of copper nanoparticles and cause a difference in the etching rate on the silicon surfaces, which eventually leads to anisotropic etching as well. The Si {100} crystal plane has a higher density of surface suspended bonds compared to the Si {111} crystal plane. Compared with the Si {111} crystal plane, the Si {100} crystal plane loses electrons more easily, so $Cu^{2+}$ can easily acquire electrons from the Si {100} crystal plane and have a much higher deposition rate on the Si {100} crystal plane than on that of the Si {111} crystal plane. This causes a difference in the etching rate between the Si {100} crystal plane and the Si {111} crystal plane. As the reaction proceeds, due to the fast etching rate of Si {100} crystalline planes, the final structure is an inverted pyramidal structure composed of four Si {111} crystalline planes [40,41].

Sticky silicon ingot glue is an epoxy resin, whose hydrolytic degradation was accelerated by $OH^-$ in the KOH solution. This reduced bonding performance. To understand the effect of KOH solution on stain removal, a metallographic microscope (MO) was used to observe changes in stain morphology at different cleaning times. Figure 9a shows an MO image of an uncleaned silicon wafer. A large amount of black material stained the surface of the silicon wafer. Figure 9b shows the stain morphology after cleaning for 1 min. The amount of stain shedding was very small, which exposed the silicon substrate. When cleaned for 3 min, the stain shedding was accelerated, and the stained area gradually shrank. Stains in the upper-left and lower-right corners were removed, and those in the middle part became sparser. After 4 min of cleaning, only a few spots remained on the wafer. At 5 min, no stains were visible under the microscope, and the stains were completely removed.

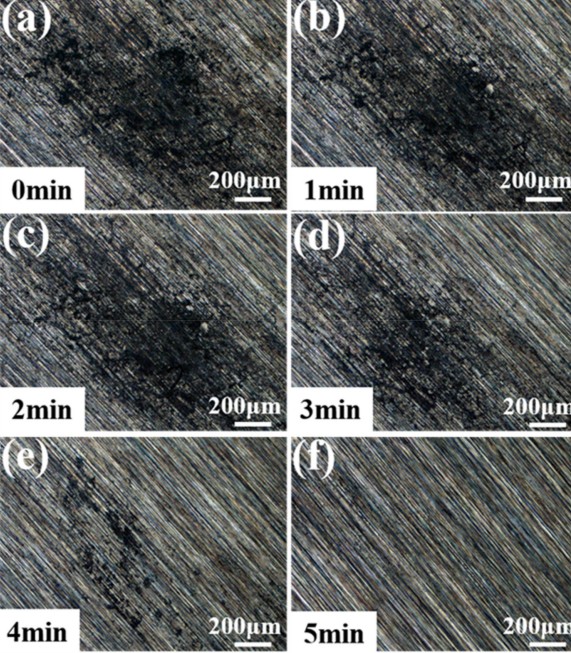

**Figure 9.** Evolution diagram of the dipped white spots with cleaning time, (**a**) MO image of white-spot-stained silicon wafer before cleaning, (**b**) cleaning for 1 min, (**c**) 2 min, (**d**) 3 min, (**e**) 4 min, (**f**) 5 min.

## 4. Conclusions

In this study, EDS and XPS were used to investigate the components and source of white spot stains on the surface of silicon wafers. The results showed that the stains contained $CaCO_3$ and $SiO_2$, which was consistent with a component analysis of the fillers of sticky silicon ingot glue. Thus, this paper inferred that the white spot stains originated from the glue residing on the diamond wire surface during the wire-cutting process. The white spot stains remained on the silicon wafer surface after the metal copper catalytic etching treatment in $HF/Cu(NO_3)_2/H_2O_2$ system was used, showing good acid corrosion resistance. The existence of the stains affected the uniform deposition of copper particles and hindered the surface texturing of silicon wafers. Finally, KOH solution combined with an ultrasonic field was proposed to remove surface stains from the silicon wafer, which were completely removed within 5 min. This study provides important guidance for the cleaning of contaminated silicon wafer surfaces, which is important in fabricating high-efficiency solar cells.

**Author Contributions:** L.M.: Data curation, Writing—original draft; X.C.: Funding acquisition, Conceptualization; C.T.: Software; S.L.: Funding acquisition, Visualization, Investigation; F.X.: Funding acquisition, Supervision, Writing—review & editing; H.L.: Funding acquisition, Supervision, Writing—review & editing; W.M.: Funding acquisition, Supervision; Y.C.: Writing—review & editing. All authors have read and agreed to the published version of the manuscript.

**Funding:** This research was funded by the National Natural Science Foundation of China (Grant No. 52274408, 51974143, 52204314, 52164050, 51904134); Major Science and Technology Projects in Yunnan Province (No. 202202AB080010, 202103AA080004, 202102AB080016, 202202AG050012); Yunnan Fundamental Research Projects (grant NO. 202201AW070014); Yunnan Provincial Rural Energy Engineering Key Laboratory (Grant No. 2022KF012), Yunnan High-level Talent Project (YNQR-GCC-2019-010) and the Program for Innovative Research Team in University of Ministry of Education of China (No. IRT_17R48).

**Institutional Review Board Statement:** Not applicable.

**Informed Consent Statement:** Not applicable.

**Data Availability Statement:** Not applicable.

**Conflicts of Interest:** The authors declare that they have no known competing financial interest or personal relationships that could have appeared to influence the work reported in this paper.

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
