# Peer review of "Effect of Silicon Wafer Surface Stains on Copper-Assisted Chemical Etching"

_metals, doi:10.3390/met13040742_

Round 1

Reviewer 1 Report

Line 45

What does it mean “dangling bonds have high energy”?

What is meant under strong activity? Do they affect e.g. transport properties?

General comment

It would be good to discuss if the KOH-ultrasound cleaning influenced Cu-assisted etching. If, e.g. pore morphology is affected, or cutting cracks may expand during sonication.

Reviewer 2 Report

In this manuscript the authors systematically investigated the composition, source, cleaning method of typical white spot stains on silicon wafer surface, representing an interesting guidance on knowledge and cleaning of silicon wafer contamination for fabricating high-efficiency solar cells.

In my opinion, the quality of the manuscript is high in terms of characterizations performed, readibility,results and perspectives, avarage in terms of process explanations and data interpretation.

I would only reccomend to improve XPS data interpretation adding quantitative information for CaCO3 stoichiometry.

A couple of further comments:

1) please check the presence of inappropriate capital letters among the manuscript (e.g. line 19, The effects - line 244, the existence)

2) paragraph 2.2, Deposition and Etching.

The composition of the etching solution and of the deposition solution is the same. Please check.

3) Figure 6. I would increase the quality of the schematic and the description in the text of the diagram of diamond wire attachment stains. 

From my point of view, for example, it is not really clear the role/movement of the green cylinders, the role/function of the plastic board in the middle of the silicon ingot and the crystal bead and the role of the salver (are the board and the salver really necessary?). Please try to improve this description.

4) Figure 7. I would specify in the caption that EDS maps in fig. c-d-e-f were taken after depostion.

Reviewer 3 Report

Dear Researchers,

Thank you for introducing such good piece of scientific work. You are clearly represent the problem. You analyzed it in a satisfied manner and finally suggested the solution as well as you proved its ability to solve the problem. 

However, the article needs minor language improvements. For example:

Try to shorten the phrases in the "Abstract" section instead of ,'s.

Please avoid inserting "We" along through the course of the article.

Section 2.2. needs to be clearer and avoid repeat phrases.  

Thank you.
